# Efficient Receptive Field Learning by Dynamic Gaussian Structure

**Evan Shelhamer    Dequan Wang    Trevor Darrell**
UC Berkeley
{shelhamer,dqwang,trevor}@cs.berkeley.edu

## Abstract

The visual world is vast and varied, but its variations divide into structured and unstructured factors. Structured factors, such as scale and orientation, admit clear theories and efficient representation design. Unstructured factors, such as what it is that makes a cat look like a cat, are too complicated to model analytically, and so require free-form representation learning. We compose structured Gaussian filters and free-form filters, optimized end-to-end, to factorize the representation for efficient yet general learning. Our experiments on dynamic structure, in which the structured filters vary with the input, equal the accuracy of dynamic inference with more degrees of freedom while improving efficiency.

## 1 Introduction

Although the visual world is varied, there is nevertheless ubiquitous structure. Free-form learned representations are structure-agnostic, making them general, but their not harnessing structure is computationally and statistically inefficient. Structured representations like steerable filtering (Freeman & Adelson, 1991; Jacobsen et al., 2016), scattering (Bruna & Mallat, 2013), and steerable networks (Cohen & Welling, 2017) efficiently express certain structures, but are constrained. We propose the semi-structured composition of Gaussian and free-form filters to blur the line between free-form and structured representations.

The effectiveness of strongly structured representations hinges on whether they encompass the true structure of the data. If not, the representation is limiting, and subject to error. At least, such is the case when structure *substitutes* for learning. In this work we *compose* structured and free-form filters and learn both end-to-end (Figure 1). The free-form parameters are not constrained by our composition for generality. The structured parameters are low-dimensional for efficiency.

We choose Gaussian structure to represent the spatial structures of scale, aspect, and orientation through covariance (Lindeberg, 1994). Optimizing these structured covariance parameters carries out a form of differentiable architecture search over receptive fields. Since this structure is low-dimensional, it is computationally efficient and could be learned from limited data.

## 2 Composing Gaussian and Free-form Filters

Our composition $f_\theta \circ g_\Sigma$ combines a free-form $f_\theta$ with a structured Gaussian $g_\Sigma$. This semi-structured composition factorizes the representation into spatial Gaussian receptive fields and free-form features. Composing filters in this fashion is a novel approach to making receptive field shape differentiable, low-dimensional, and decoupled from the number of parameters.

The structure of a Gaussian is controlled by its covariance $\Sigma$, which for a spatial 2D Gaussian is $\begin{bmatrix} \sigma_y^2 & \rho \\ \rho & \sigma_x^2 \end{bmatrix}$ with elements $\sigma_y^2$, $\sigma_x^2$ for the y, x coordinates and $\rho$ for their correlation. The standard, isotropic Gaussian has identity covariance $\begin{bmatrix} 1 & 0 \\ 0 & 1 \end{bmatrix}$. Covariances come in families with progressively richer structure: spherical has one parameter for scale, diagonal has two parameters for scale and aspect, and full has three parameters for scale, aspect, and orientation/slant.

**Covariance Parameterization & Optimization** The covariance $\Sigma$ is symmetric positive definite, so it requires proper parameterization for optimization. The log-Cholesky parameterization (Pin-

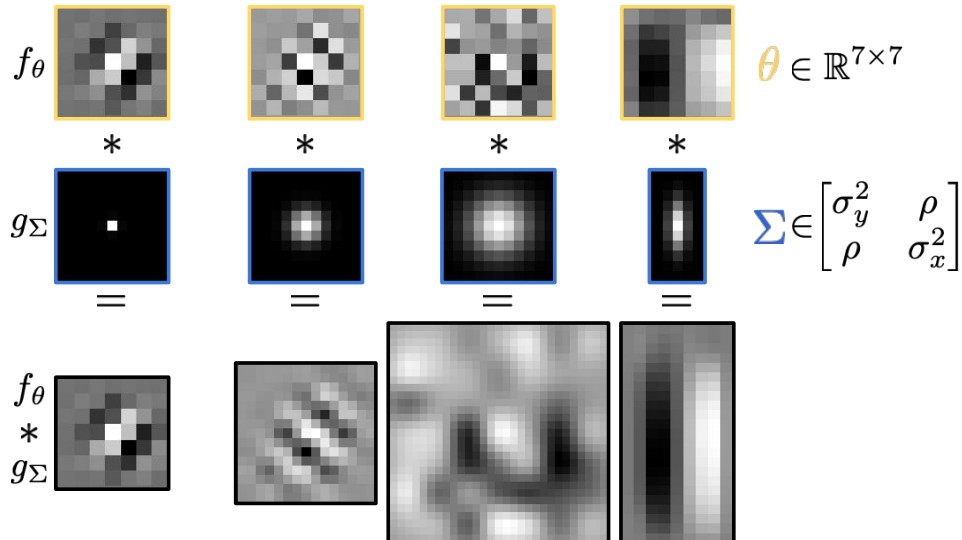

Figure 1: We compose free-form filters $f_\theta$ and structured Gaussian filters $g_\Sigma$ by convolution $*$ to define a more general family of semi-structured filters than can be learned by either alone. Our composition makes receptive field scale, aspect, and orientation differentiable in a low-dimensional parameterization for efficient end-to-end learning.

heiro & Bates, 1996) is a good choice for iterative optimization because it is simple and quick to compute: $\Sigma = U'U$ for upper-triangular $U$ with positive diagonal. We can keep the diagonal positive by storing its log, hence *log*-Cholesky, and exponentiating it when forming $\Sigma$.

**Composing with Convolution and Covariance** The computation of our composition reduces to convolution, and so it inherits the efficiency of aggressively tuned convolution implementations. Convolution is associative, so compositionally filtering an input $I$ decomposes into two steps of convolution by

$$I * (g_\Sigma * f_\theta) = I * g_\Sigma * f_\theta. \tag{1}$$

This decomposition has computational advantages. The Gaussian step can be done by specialized filtering that harnesses separability, cascade smoothing, and other Gaussian structure. Memory can be spared by only keeping the covariance parameters and recreating the Gaussian filters as needed (which is quick, although it is a space-time tradeoff). Each compositional filter can always be explicitly formed by $g_\Sigma * f_\theta$ for visualization (see Figure 1) or other analysis.

Both $\theta$ and $\Sigma$ are differentiable for end-to-end learning.

**Dynamic Gaussian Structure** Semi-structured composition can learn a rich family of receptive fields, but visual structure is richer still, because structure locally varies while our filters are fixed. Even a single image contains variations in scale and orientation, so one-size-and-shape-fits-all structure is suboptimal. *Dynamic* inference replaces static, global parameters with dynamic, local parameters that are inferred from the input to adapt to these variations. Composing with structure by convolution cannot locally adapt, since the filters are constant across the image. We can nevertheless extend our composition to dynamic structure by representing local covariances and instantiating local Gaussians accordingly. Our composition makes dynamic inference efficient by decoupling low-dimensional, Gaussian structure from high-dimensional, free-form filters.

There are two routes to dynamic Gaussian structure: local filtering and deformable sampling. Local filtering has a different filter kernel for each position, as done by dynamic filter networks (De Brabandere et al., 2016). This ensures exact filtering for dynamic Gaussians, but is too computationally demanding for large-scale recognition networks. Deformable sampling adjusts the position of filter taps by arbitrary offsets, as done by deformable convolution (Dai et al., 2017). We exploit deformable sampling to dynamically form sparse approximations of Gaussians.

We constrain deformable sampling to Gaussian structure by setting the sampling points through covariance. Figure 2 illustrates these Gaussian deformations. We relate the default deformation to

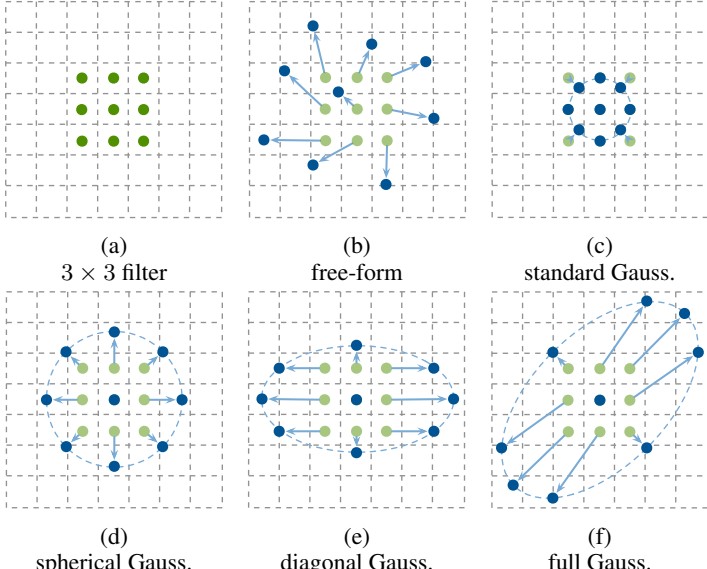

|        |        |        |
|--------|--------|--------|
| (a) $3 \times 3$ filter | (b) free-form | (c) standard Gauss. |
| (d) spherical Gauss. | (e) diagonal Gauss. | (f) full Gauss. |

Figure 2: Gaussian deformation (c-f) structures dynamic receptive fields by controlling the sampling points (blue) through the covariance. The low-dimensionality of covariance is more efficient than free-form deformation (b) for learning and inference. Although it is less general, it still expresses a variety of shapes.

the standard Gaussian by placing one point at the origin and circling it with a ring of eight points on the unit circle at equal distances and angles. We consider the same progression of spherical, diagonal, and full covariance for dynamic structure. This low-dimensional structure differs from the high degrees of freedom in a dynamic filter network, which sets free-form filter parameters, and deformable convolution, which sets free-form offsets. In this way our semi-structured composition requires only a small, constant number of covariance parameters independent of the sampling resolution and the kernel size $k$, while deformable convolution has constant resolution and requires $2k^2$ offset parameters for a $k \times k$ filter.

To infer the local covariances, we follow the deformable approach (Dai et al., 2017), and learn a convolutional regressor for each dynamic filtering step. The regressor, which is simply a convolution layer, first infers the covariances which then determine the dynamic filtering that follows. The low-dimensional structure of our dynamic parameters makes this regression more efficient than free-form deformation, as it only has three outputs for each full covariance, or even just one for each spherical covariance. Since the covariance is differentiable, the regression is learned end-to-end from the task loss without further supervision.

## 3 EXPERIMENTS

We experiment on semantic segmentation of CityScapes (Cordts et al., 2016), a challenging dataset of varied urban scenes captured by a car-mounted camera. We score results by the common intersection-over-union metric (IU). We choose the fully convolutional DRN-A (Yu et al., 2017) as our base architecture. We choose deformable convolution (Dai et al., 2017) as strong baseline for local, dynamic inference without structure. We train all methods with the same optimization settings for fair comparison.

Note that the backbone is an aggressively-tuned architecture which required significant model search and engineering effort. Our composition is still able to deliver improvement through learning without further engineering.

We compare our static composition and our Gaussian deformation with free-form deformation in Table 1. We augment the last, output stage with our composition and optimize end-to-end. Static

| Cityscapes Validation | | | |
|---|---|---|---|
| method | dyn.? | no. dyn. params | IU |
| DRN-A (Yu et al., 2017) | | - | 72.4 |
| + Gaussian by Convolution (ours) | | - | 73.5 |
| + Gaussian by Deformation (ours) | ✓ | **1** | **76.6** |
| + Free-form Deformation (Dai et al., 2017) | ✓ | $2k^2$ | **76.6** |
| Cityscapes Test | | | |
| DRN-A (Yu et al., 2017) | | - | 71.2 |
| + Gauss. Deformation (ours) | ✓ | **1** | **74.3** |
| + Free-form Deformation (Dai et al., 2017) | ✓ | $2k^2$ | 73.6 |

Table 1: Dynamic Gaussian deformation reduces parameters, improves computational efficiency, and rivals the accuracy of free-form deformation. Even restricting the deformation to scale by spherical covariance suffices to equal the free-form accuracy.

composition by convolution improves on the backbone by 1 point while dynamic Gaussian deformation gives a further $+3$ points.

Controlling deformable convolution by Gaussian structure improves efficiency while preserving accuracy to within one point. While free-form deformations are more general in principle, in practice there is a penalty in efficiency. Recall that the size of our structured parameterization is independent of the free-form filter size. On the other hand the original, unstructured deformable convolution requires $2k^2$ parameters for a $k \times k$ filter.

Our results show that making scale dynamic through spherical covariance suffices to achieve equal (or near equal) accuracy. Scale is perhaps the most ubiquitous transformation in the distribution of natural images, so scale modeling might suffice to handle many transformations. Our low-dimensional parameterization, needing only one scale parameter at the extreme, can be efficiently optimized on limited data.

## 4 Conclusion

Composing structured Gaussian filters and free-form filters enables efficient receptive field learning. This kind of factorization points to a reconciliation of structure and learning, through which known visual structure is respected and unknown visual detail is learned freely.

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
