# OpenReview forum: "Efficient Receptive Field Learning by Dynamic Gaussian Structure"
_ICLR.cc/2019/Workshop/LLD — LLD 2019_

### Official Review · AnonReviewer1 · 2019-04-07
**The paper should be made more precise and conclusive**

**Rating:** 2
**Confidence:** 2

**Review:**

This paper proposes a structured convolution operator to model deformations of local regions of an image. The deformation field is parameterized by a Gaussian function. The advantage of this approach is that it significantly reduced the number of parameters. The result on image segmentation shows that it can achieve good accuracy and is more efficient.

The reviewer is unable to justify the advantage of the idea in the regime of small data samples.
The paper is somewhere hard to follow,  due to the imprecise expression like “learned end-to-end”, “Static Composition”, etc. Is it to learn both parameters theta and Sigma, or just Sigma, or just theta? The result in Table 1 would be more clear if these were made precise. The formula (1), is it correct or not? Since convolution with g_Sigma involves bi-linear interpolation, the convolution does not seem to be done on the grid of I and f_theta.

Overall, the idea is interesting, but the advantage of the approach to address small-sample problem is not conclusive enough. The experiments would be more convincing if it could be tested on other problems such as classification, detection.

---

### Official Review · AnonReviewer2 · 2019-04-08
**Interesting idea with nice results**

**Rating:** 4
**Confidence:** 2

**Review:**

This paper proposes a method that offers middle ground between deterministic structured filter representations and learnable free-form representations. The idea described in the paper amounts to  learning the parameters of a fixed spatial structure (i.e., covariance matrix of a gaussian distribution), as well as the parameters of a free form filter. The combination of the two is expressed through a convolution composition. The advantage of the proposed method is the lower number of parameters needed, to achieve similar performance to e.g., deformable sampling methods; the lower number of parameters can in-turn be useful in scenaria where data is scarce.



One question I have is in regards to the "static composition" variant, which is not clearly described in the paper. Results using that variant are also listed only for the Cityscapes validation set.

---

### Decision · Program_Chairs · 2019-04-09
**Acceptance Decision**

Accept